# The Association between Message Framing and Intention to Vaccinate Predictive of Hepatitis A Vaccine Uptake

**DOI:** 10.3390/ijerph21020207

**Published:** 2024-02-10

**Authors:** Nora Satybaldiyeva, Lourdes S. Martinez, Brittany Cooper, Eyal Oren

**Affiliations:** 1Division of Epidemiology and Biostatistics, School of Public Health, San Diego State University, San Diego, CA 92182, USA; ksatybaldiyeva3221@sdsu.edu (N.S.); brittanycooper3@gmail.com (B.C.); 2School of Communication, San Diego State University, San Diego, CA 92182, USA; lsmartinez@sdsu.edu

**Keywords:** vaccine, intention, hepatitis A virus, vaccination intention, random allocation

## Abstract

As ongoing, sporadic outbreaks of hepatitis A virus (HAV) infections present public health challenges, it is critical to understand public perceptions about HAV, especially regarding vaccination. This study examines whether message framing changes the intention to vaccinate against HAV and self-reported vaccine behavior. Using a randomized controlled trial (N = 472) in February 2019 via Amazon Mechanical Turk, participants were randomized to one of four HAV vaccination message groups or a no-message control group. The message groups varied in their emphasis on the nature of outcomes (gain versus loss) and for whom (individual versus collective). The message frames were compared by intention to vaccinate, differences in message characteristics, and behavioral determinants. There was no difference in intention to vaccinate between gain- versus loss-framed messages (MD = 0.1, 95% CI = −0.1, 0.3) and individual- versus collective-framed messages (MD = 0.1, 95% CI = −0.1, 0.3). The intention to vaccinate against HAV in the no-message control group was very similar to that in the message groups. However, gain-framed messages were rated more positively in valence than loss-framed messages (MD = −0.5, 95% CI = −0.7, −0.3), which may be helpful for cultivating a positive public perception of HAV vaccination. The study also highlights the importance of comparing message frames to a no-message control in designing health communication messaging promoting HAV vaccination.

## 1. Introduction

Hepatitis A virus (HAV) is the most common cause of viral hepatitis worldwide and is primarily transmitted fecal-orally after close contact with an infected person [1]. While the rates of HAV had declined by approximately 95% from 1996 to 2011 in the United States (US), from 2016 to 2018, approximately 15,000 new HAV infections were reported in US states and territories, indicating a recent increase in transmission [2]. Since 2017, the vast majority of these reports have been related to multiple outbreaks of infections among people reporting drug use or homelessness [3]. The most recent outbreak of HAV in the US, which occurred between March and August of 2022, was linked to fresh organic strawberries [4]. While food-borne outbreaks of HAV are less common, they occur when uninfected and unvaccinated individuals consume food that has been handled by someone with HAV [5]. A large outbreak of HAV in San Diego was officially declared a local public health emergency by the San Diego Public Health Officer on 1 September 2017. The outbreak, which progressed and spread to nearby areas, including Los Angeles and Santa Cruz, was notable both in its severity and its rapid spread, with almost 600 cases, 20 deaths, and 400 hospitalizations in San Diego County [6]. In addition, since the outbreak, numerous other states have reported cases of HAV, some of which were linked to the initial outbreak in San Diego [3].

During such outbreak situations, a public health strategy must be coordinated to ensure that the general public is adequately informed about risks and appropriate behavioral responses. It is thus critical to understand public perceptions about HAV in general and especially regarding vaccination in response to recent outbreaks in order to frame appropriate risk messaging. Specifically tailored messages can encourage the uptake of vaccinations and preventive measures for HAV, in addition to reducing the challenges that accompany vaccination campaigns, such as vaccine hesitancy. Vaccine hesitancy, the delay or refusal of vaccination due to misconceptions or concerns about the available vaccine [7], is especially important with regard to the recent environment surrounding coronavirus disease 2019 (COVID-19). Surveys examining sentiments around COVID-19 vaccination have exposed new levels of volatility around vaccine hesitancy, often powered by social and digital media platforms [8]. Therefore, vaccine hesitancy represents a crucial time of vulnerability and opportunity for public health response and vaccine communication measures. While traditional vaccines, such as the HAV vaccine, and new vaccines, such as the COVID-19 vaccine, share the common goal of preventing infectious diseases, differences in their development, public perception, and the influence of vaccine hesitancy necessitate tailored communication strategies, such as message framing, to ensure widespread acceptance and participation in vaccination programs.

Message framing has been frequently assessed in vaccine communication research [9]. One of the most common message frames to receive scholarly attention is gain/loss [10,11]. Gain/loss messages are framed according to whether they focus on losses or gains (with informational equivalence held constant) and generate different framing effects. According to the prospect theory, individuals presented with a message that emphasizes losses will respond with greater risk-averse behavior [12,13]. In contrast, individuals presented with a message highlighting gains will respond with greater risk-seeking behavior. Gain/loss messages can also be crafted using goal framing, which emphasizes the consequences of the actions described in these messages. Messages emphasize either loss or gain consequences resulting from particular actions [14], which have been used in several prior vaccine communication studies [9]. However, systematic reviews of the literature suggest mixed effects regarding whether one message framing is more effective than the other [9,11].

Furthermore, it is unclear how the effectiveness of gain/loss messages may improve behavioral responses depending on the emphasis on for whom consequences (self vs. collective). Although previous studies have compared the consequences of actions to themselves or others [15,16] and suggested that vaccine messages that emphasize a collective loss (compared to an individual loss) may be more effective in promoting pro-vaccine attitudes and intentions [17], more research is needed in this area. Therefore, we examined whether messages emphasizing collective loss are more likely to be effective in promoting HAV vaccination intention when compared to messages emphasizing individual loss.

Finally, given the recent politics surrounding COVID-19 vaccines [18,19], it is important to determine the potential for boomerang effects of vaccine messages. A boomerang effect occurs when a message produces the complete opposite response in the audience to what was intended by the communicator [20,21]. Despite this potential for unintended negative effects on audiences, not all vaccine communication studies assess vaccine framing compared to a control group, which could present challenges in detecting when vaccine messages have the potential to backfire. To help address this gap, we included a control group in our design.

Our primary objective was to examine whether message framing changes self-reported intention to receive the HAV vaccine and behavioral determinants of intention. We hypothesized that loss frames are more powerful than gain frames for participants’ intention to vaccinate against HAV. Our expectation was rooted in research suggesting the existence of a general negativity bias among individuals, resulting in a greater prioritization of negative information over positive information [22,23]. Furthermore, we included a no-message control group in our study design to assess whether not receiving any message influences self-reported intention to receive the HAV vaccine.

## 2. Materials and Methods

### 2.1. Study Design

We performed a 2 (gain vs. loss) × 2 (individual vs. collective) between-subjects message-testing experiment with a control group using the Qualtrics platform and Amazon Mechanical Turk (MTurk) from 11 February to 17 March 2019 (when the maximum number of respondents was reached). MTurk is an online platform that utilizes human volunteers to complete human intelligence tasks (HITs) [24]. The use of the MTurk platform for behavioral research has been growing over the past several years due to the increasing diversity of volunteers and the ability to recruit large samples of participants in a short period of time [25]. The study protocol was reviewed by the University Institutional Review Board and deemed exempt on 25 January 2019.

### 2.2. Study Participants

Participants were recruited under MTurk restrictions that they are current US residents and have at least a 95% task approval rate for previous HITs in order to increase the likelihood of task completion. All eligible participants were provided with a brief study description, and those who were interested were directed anonymously to the Qualtrics platform. Participants were then given more information about the survey and asked whether they resided in San Diego County and whether they were at least 18 years old. Eligible participants who electronically consented to the study were randomized to one of four message frames or a control group and proceeded to answer follow-up survey questions.

### 2.3. Message Frames

Participants were randomized to one of five groups: collective-gain-framed message, collective-loss-framed message, individual-gain-framed message, individual-loss-framed message, or a control group that received no message. The four message groups varied in their emphasis on the nature of outcomes (gain versus loss) and for whom (individual versus collective). Message frames were designed to resemble a format that members of the general public are likely to encounter when consuming news information. Specifically, the messages were designed to resemble a Twitter post shared by a local health group (see Appendix A). This format was appropriate as most US adults report getting at least some of their news from social media [26,27], with Twitter representing a popular social media platform that users consult for breaking news [28].

### 2.4. Instrument

Participants answered questions about sociodemographic factors and prior HAV vaccination before random assignment to a message frame or a no-message control group. Participants randomized to one of the four message frame groups answered questions capturing message-related characteristics. Lastly, all participants answered items related to behavioral determinants of intention to vaccinate against HAV.

Participants self-reported their age, sex, ethnicity, race, and education. Three vaccine behavior questions were asked about belonging to a group for whom the vaccine has been recommended, obtaining the HAV vaccine when recommended, and being in close contact with someone with HAV. We also assessed four message-related characteristics: valence, credibility, likability, and perceived effectiveness. Message valence was measured with a single item that asked how positively or negatively the participant viewed the message [29]. Message credibility consisted of 3 items that asked whether the participant would describe the message they received as (1) believable, (2) accurate, and (3) authentic [30]. Message likability was measured using 6 items on a 5-point scale that asked participants to rate the degree to which the message they received was (1) enjoyable, (2) lively, (3) boring, (4) pleasing, (5) helpful, and (6) interesting [31]. The item that assessed the extent to which the message was boring (item 3) was reverse coded to be summed with the other items. Perceived effectiveness was assessed using 2 items measured on a 5-point scale that asked how likely the message would (1) persuade and (2) convince someone to receive the HAV vaccine [32].

The behavioral determinants assessed by the survey consisted of items from the health belief model (HBM). The HBM items assessed perceived susceptibility, severity, benefits, barriers, and cues to action adapted from prior research [33]. Perceived susceptibility consisted of two constructs using two separate items: perceived need and perceived risk. Perceived benefits consisted of a sum of two items that assessed whether the HAV vaccine was (1) preventive and (2) protective. Perceived barriers comprised a sum of 4 items that measured (1) danger to health, (2) lack of access, (3) lack of money, and (4) lack of time necessary to get the vaccine. The cue to action was a sum of 5 items that examined whether the participant (1) planned to or had already received the HAV vaccine or would get the vaccine if (2) it was recommended, (3) encouraged by a significant other, (4) encouraged by someone online, or (5) offered for free at their workplace. All HBM questions were measured on a 5-point scale with the additional option of “not applicable”. Participants who selected “not applicable” (sample sizes ranging from 37 to 68) were excluded from the analyses for those respective constructs.

The dependent variable, HAV vaccine intention, was measured on a 5-point scale that assessed agreement with the statement “I intend to get the Hepatitis A shot when it is recommended”, derived from the theory of planned behavior [34].

### 2.5. Data Analysis

Descriptive characteristics were obtained for all five groups and compared to ensure effective randomization. Questions measuring the same message characteristics and behavioral determinants were assessed for internal consistency (Cronbach’s alpha > 0.6). The items summed to correspond to the different constructs (message credibility, likability, perceived effectiveness, norms, benefits, barriers, and cue to action) were subjected to inter-item correlation analysis. Gain versus loss and collective versus individual message frames were compared for the main outcome of interest (intention to vaccinate), the different message characteristics, and behavioral determinants using one-tailed *t*-tests. The mean of the score for each characteristic was analyzed as a mean difference (MD) between the two respective message frame groups. Stratified analyses were performed to examine whether intention to vaccinate varied by self-reported sex and age. An additional stratified analysis was conducted in order to observe whether responses to the TPB and HBM questions varied by receipt of the hepatitis A vaccination within the past year at the time of survey completion. A manipulation check was performed to assess the effectiveness of the different message frames. All statistical analyses were conducted using SAS version 9.4.

We conducted a post-hoc power analysis to determine whether the existing sample size had adequate study power for all the proposed analytical aims. Assuming an effect size (Cohen’s d) = 0.5, α = 0.05, a sample size of 90 for group 1, and a sample size of 92 for group 2, we used a *t*-test to compute the achieved power for the difference in means between two independent groups. Our computations noted achieving 95.7% power. All power calculations were conducted using G*Power version 3.1 (Heinrich Heine University Düsseldorf, Düsseldorf, Germany)

## 3. Results

### 3.1. Study Sample

A total of 1701 people expressed interest in the study. Of those, 1181 were excluded because they did not provide consent, did not meet the eligibility criteria, declined to participate, or did not proceed with the survey. A total of 520 participants were eligible and randomized. Of the 520 eligible participants, 20 were excluded for not answering all the survey questions, and 28 were excluded for incorrectly answering over half of the attention-check questions that they encountered in the survey. The analytic sample used for this analysis consisted of 472 people who completed the survey (Figure 1). Of the 472 participants, 92 (19.5%) were randomized to the individual-gain-framed message, 99 (21.0%) to the individual-loss-framed message, 90 (19.1%) to the collective-gain-framed message, 98 (20.8%) to the collective-loss-framed message, and 93 (19.7%) to the no-message control group.

### 3.2. Participant Characteristics

Most of the participants were between the ages of 25 and 44 (70.6%), White (63.4%), not Hispanic or Latino (69.9%), and had an associate degree or higher (73.1%) (Table 1). About half of all participants were female (51.3%). About 43% of participants belonged to a group for whom the hepatitis A vaccine has been recommended, 58% received the hepatitis A vaccine when it was recommended, and 34% had been in contact with someone with hepatitis A. Most participants had a strong intention to get the HAV vaccine (μ = 3.96, SD = 0.96).

### 3.3. Intention to Vaccinate

We found that participants in the loss-framed message group (μ = 4.0) did not have a higher intention to vaccinate than those in the gain-framed message group (μ = 3.9, MD = 0.1, 95% CI = −0.1, 0.3). Similarly, while we hypothesized that there would be an advantage in using a collective-framed message compared to an individual-framed message, we found no significant difference in intention to vaccinate when comparing collective (μ = 4.0) to individual (μ = 3.9, MD = 0.1, 95% CI = −0.1, 0.3) message groups. Lastly, there was no advantage to using any of the four message frames compared to the no-message control group (μ = 4.0).

### 3.4. Message-Related Characteristics

As seen in Table 2, gain-framed messages (μ = 3.9) were rated more positively in valence than loss-framed messages (μ = 3.4) (MD = −0.5, 95% CI = −0.7, −0.3). Otherwise, there were no noticeable differences between gain-framed and loss-framed messages in terms of message credibility (MD = 0.3, 95% CI = −0.3, 0.9), likability (MD = 0.4, 95% CI = −0.4, 1.3), and perceived effectiveness (MD = 0.2, 95% CI = −0.3, 0.7). There was no difference between collective-framed and individual-framed messages for message valence (MD = 0.1, 95% CI = −0.1, 0.3), credibility (MD = −0.4, 95% CI = −1.0, 0.2), likability (MD = −0.3, 95% CI = −1.2, 0.5), and perceived effectiveness (MD = −0.3, 95% CI = −0.8, 0.2).

### 3.5. Health Belief Model

For both items assessing susceptibility, gain-framed messages generated lower levels of need and risk (μ = 2.9 and μ = 3.0, respectively) compared to loss-framed messages (μ = 3.2 and μ = 3.2, respectively) and the control group (μ = 3.4 and μ = 3.3, respectively). Gain-framed messages (μ = 8.1) were also more likely to lead to a reduced sense of benefit of getting the HAV vaccine when compared to the loss-framed messages (μ = 8.4, MD = 0.3, 95% CI = −0.1, 0.6) but not when compared to the control group (μ = 8.2, MD = 0.0, 95% CI = −0.5, 0.4). On the other hand, gain-framed messages (μ = 9.5) significantly reduced the perception of barriers when compared to the control group (μ = 10.7, MD = −1.2, 95% CI = −2.3, 0.0) but not when compared to the loss-framed message group (μ = 9.3, MD = −0.2, 95% CI = −1.1, 0.7). Lastly, gain-framed messages (μ = 17.8) reduced cue to action significantly when compared to loss-framed messages (μ = 18.9, MD = 1.1, 95% CI = 0.2, 2.0) but not when compared to the control group (μ = 18.3, MD = −0.4, 95% CI = −1.6, 0.7).

As shown in Table 3, across the different constructs of the HBM, we found no difference between collective-framed and individual-framed messages regarding risk (MD = −0.1, 95% CI = −0.4, 0.1), benefits (MD = −0.1, 95% CI = −0.5, 0.3), or cue to action (MD = 0.4, 95% CI = −0.5, 1.3). However, collective-framed messages (μ = 2.9) reduced the perception of need when compared to the control group (μ = 3.4, MD = −0.5, 95% CI = −0.8, −0.1) but not when compared to the individual-framed message group (μ = 3.1, MD = −0.2, 95% CI = −0.5, 0.1). Additionally, collective-framed messages (μ = 9.1) significantly reduced the perception of barriers when compared to the control group (μ = 10.7, MD = −1.6, 95% CI =−2.7, −0.5) but not when compared to the individual-framed message group (μ = 9.8, MD = −0.7, 95% CI = −1.6, 0.2).

Our reliability analyses indicated that all items measuring the specific constructs of the HBM, except for the items measuring perceived benefits, showed a high correlation with each other (Cronbach’s alphas > 0.7, Table 4). A more detailed list of the constructs can be found in the Appendix A. After narrowing down the items measuring perceived benefits, the perceived benefits construct showed an acceptable level (Cronbach’s alpha > 0.5) of internal consistency.

### 3.6. Stratified Analyses

Intention to vaccinate against HAV did not vary by age group (F = 0.4, *p* = 0.7) or whether the participant was male (μ = 3.9) or female (μ = 4.0, MD = 0.1, 95% CI = −0.2, 0.1). Within the entire sample, intention to vaccinate against HAV was higher among those with a high perceived risk of HAV (μ = 4.1) compared to those with a low perceived risk (μ = 3.6, MD = −0.5, 95% CI = −0.7, −0.3). There were no significant differences in intention to vaccinate by message frames between individuals with a high perceived risk of HAV and those with a low perceived risk of HAV.

## 4. Discussion

In this study, gain-framed messages did not change self-reported intention to obtain the HAV vaccine compared to loss-framed messages and the no-message control group. Additionally, intention did not differ by message appeal (collective versus individual) compared to the no-message control group. None of the four message frames were more effective than the no-message group in encouraging greater behavioral intentions to obtain HAV vaccination. These findings highlight the importance of comparing message frames to a no-message control group before developing health communication campaigns promoting HAV vaccination as efforts to emphasize other message factors may be more impactful for promoting HAV vaccine intention.

As expected, we found that gain-framed messages about HAV vaccination were viewed more positively than loss-framed messages. These results support using gain-framed messages to encourage voluntary vaccinations, such as the HAV vaccine, over messages with negative valence framing, which have been shown to produce greater support for mandatory vaccinations [35]. Additionally, results showed that gain-framed and collective-framed messages reduced barrier perceptions regarding receiving HAV vaccination compared to the no-message control group. Therefore, it may be beneficial to use gain-framed messages with a collective appeal when promoting HAV vaccinations among populations with high perceived barriers, such as individuals without health insurance.

Our results align with previous research around vaccine communication. For example, one systematic review found no significant difference in the persuasiveness of gain- and loss-framed appeals for encouraging vaccination [11]. Another systematic review found that only one study compared the effectiveness of messages with individual appeals to those with collective appeals [9]. Furthermore, most previous studies examining message framing in vaccine communication did not compare message frames to a control (no-message) group [9]. Among nine interventions examining human papillomavirus (HPV) vaccine uptake, only one study found that the intervention group increased uptake compared to the control condition [36]. More recently, in a comparison of loss versus gain and individual versus collective message frames on COVID-19 vaccination, the researchers did not find any difference in vaccine attitudes or intentions [37]. Although prior work has looked at vaccine intention for diseases such as HPV, influenza, and COVID-19, this is the first study to date that has examined the effects of message framing in combination with a control group on HAV vaccine intention [37,38,39]. Among the limited existing HAV vaccine intention work, most have focused on sociodemographic correlates of intention and have not examined the effect of message frames or a no-message control group [40,41,42].

The results of our study have implications for future theorizing and testing of message framing. For framing theory, our findings suggest that examining more distal variables, such as perceived message valence and perceived barriers to HAV vaccination, which may contribute to more proximal determinants of intention (attitude and self-efficacy, respectively), may help illuminate potential framing effects that could be partially mediated through these determinants (but whose effects decompose before reaching intention). Our study offers evidence that these distal variables may deserve more theoretical attention in future research. These results highlight the need to revisit the application of message framing to vaccination messaging research. While numerous studies have examined the effect of message framing on COVID-19 vaccination, none utilized a no-message control group as a comparison [37,43,44,45]. This study also provides practical implications for vaccine intention beyond HAV and can inform future effective vaccine communication efforts. The implication for creating targeted messages to encourage vaccination intention is that whether the messages are gain/loss-framed or individual/collective-framed should not be of primary concern. Our results may be useful for healthcare providers because they show that discussion of individual or collective benefits seems to neither add value to, nor interfere with, vaccination intentions. However, care should be taken to avoid assuming that any message framing can be appropriate for vaccine communication or that message framing can be sufficient to overcome known challenges in effectively communicating vaccine information [9].

Our study has several strengths and limitations. The results of our study are limited to one context (HAV vaccination) and may not be generalized to other vaccine messages or other health contexts. Second, the experimental nature of the study may not completely reflect real-world conditions in which members of the public encounter vaccine messages. Third, the use of MTurk for research poses many challenges, such as participant self-misrepresentation to meet study eligibility criteria, which may have influenced our findings [46]. Fourth, the majority of the study sample was young, White, and had a high level of education compared to the general US population. Therefore, more research is needed to examine the effects of these message frames among other populations vulnerable to HAV outbreaks that may not be available for research via MTurk, such as older adults [47]. One major strength is the randomized controlled trial study design of our message testing experiment, which consisted of a 2 × 2 between-subjects design with a control group. Our use of a control group without a message allowed us to make comparisons across conditions previously unexamined by past research. Another benefit of using the MTurk platform is the high response and completion rate as only 20 participants (3.8%) did not answer all the survey questions.

## 5. Conclusions

This study adds to the literature suggesting no advantage of gain-framed messaging over loss-framed messaging or collective-framed over individual-framed messaging for HAV vaccination intention. More research is needed to examine which message frames, compared to a control group, are most strongly associated with HAV or other vaccine uptake. Such associations may vary depending on different geographical locations, cities, and states in the US, especially those with existing HAV vaccination disparities among local populations. Future research may also consider examining the use of different modalities and other forms of social media in the delivery of HAV-framed messages that may affect HAV vaccination. Similarly, it would be important to study the effects of these messages among other populations vulnerable to HAV outbreaks. Lastly, more work is needed to explore factors mediating and/or moderating the observed associations between different message frames on vaccination intention and vaccine uptake.

## Figures and Tables

**Figure 1 ijerph-21-00207-f001:**
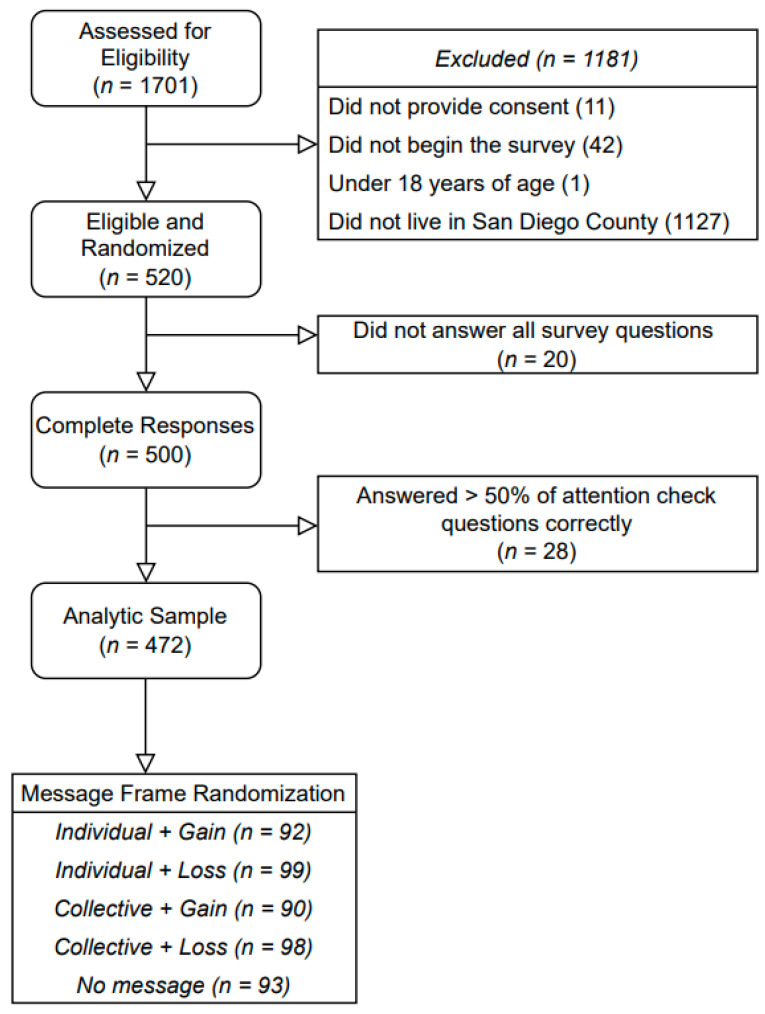
Flow diagram depicting the derivation of the analytic study sample.

**Table 1 ijerph-21-00207-t001:** Participant characteristics: overall and by message frame randomization group.

Participant Characteristics	Total N (%)	Gain + Individual n (%)	Gain + Collective n (%)	Loss + Individual n (%)	Loss + Collective n (%)	Control n (%)
	472	92 (19.5)	90 (19.1)	99 (20.1)	98 (20.8)	93 (19.7)
Age						
18–24 years	97 (20.6)	21 (22.8)	19 (21.1)	23 (23.2)	15 (15.3)	19 (20.4)
25–44 years	333 (70.6)	63 (68.5)	59 (65.6)	70 (70.7)	76 (77.6)	65 (69.9)
45 years or greater	38 (8.1)	8 (8.7)	12 (13.3)	6 (6.1)	7 (7.1)	9 (9.7)
Sex						
Male	227 (48.1)	40 (43.5)	44 (48.9)	48 (48.5)	51 (52.0)	44 (47.3)
Female	242 (51.3)	52 (56.5)	45 (50.0)	51 (51.5)	45 (45.9)	49 (52.7)
Other	3 (0.6)	0	1 (1.1)	0	2 (2.0)	0
Ethnicity						
Hispanic or Latino	144 (30.5)	23 (25.0)	30 (33.3)	28 (28.3)	35 (35.7)	28 (30.1)
Not Hispanic or Latino	328 (69.5)	69 (75.0)	60 (66.7)	71 (71.7)	63 (64.3)	65 (69.9)
Race						
White	299 (63.4)	65 (70.7)	60 (66.7)	54 (54.6)	63 (64.3)	57 (61.3)
Black	35 (7.4)	4 (4.4)	6 (6.7)	12 (12.1)	6 (6.1)	7 (7.5)
Asian	66 (14.0)	15 (16.3)	11 (12.2)	14 (14.1)	10 (10.2)	16 (17.2)
Other ^a^	60 (12.7)	8 (8.7)	13 (14.4)	19 (19.2)	19 (19.4)	13 (14.0)
Education						
Less than high school	2 (0.4)	0	0	0	0	2 (2.2)
High school/some college	125 (26.5)	24 (26.1)	31 (34.4)	24 (24.2)	26 (26.5)	20 (21.5)
Associate degree or higher	345 (73.1)	68 (73.9)	59 (65.6)	75 (75.8)	72 (73.7)	71 (76.3)
Belong to a group for whom HAV has been recommended						
Yes	205 (43.4)	38 (41.3)	34 (37.8)	44 (44.4)	42 (42.9)	47 (50.5)
No	153 (32.4)	28 (30.4)	32 (35.6)	29 (29.3)	35 (35.7)	29 (31.2)
Don’t Know	114 (24.2)	26 (28.3)	24 (26.7)	26 (26.3)	21 (21.4)	17 (18.3)
Got HAV vaccine when it was recommended						
Yes	274 (58.1)	55 (59.8)	47 (52.2)	61 (61.6)	54 (55.1)	57 (61.3)
No	120 (25.4)	20 (21.7)	25 (27.8)	18 (18.2)	30 (30.6)	27 (29.0)
Don’t Know	78 (16.5)	17 (18.5)	18 (20.0)	20 (20.2)	14 (14.3)	9 (9.7)
In close contact with someone with HAV						
Yes	159 (33.7)	31 (33.7)	31 (34.4)	29 (29.3)	31 (31.6)	37 (39.8)
No	155 (32.8)	26 (28.3)	28 (31.1)	36 (36.4)	34 (34.7)	31 (33.3)
Don’t Know	158 (33.5)	35 (38.0)	31 (34.4)	34 (34.3)	33 (33.7)	25 (26.9)
Intention ^b^, Mean (SD)	3.96 (0.96)	3.80 (1.0)	3.98 (1.0)	4.05 (0.9)	3.99 (1.0)	3.95 (1.0)

Note. Percentages may not add up to 100% due to rounding; HAV = hepatitis A virus. ^a^ Other races include Native American/Alaskan Native, Native Hawaiian/Pacific Islander, and others. ^b^ Intention was measured on a 5-point scale ranging from strongly disagree (1) to strongly agree (5).

**Table 2 ijerph-21-00207-t002:** Mean message-related responses regarding HAV vaccine intention across different message frame groups.

Message-Related Characteristics	Loss Frame	Gain Frame	Loss vs. Gain MD (95% CI)	Collective Frame	Individual Frame	Collective vs. Individual MD (95% CI)
Valence ^a^	3.4 (1.2)	3.9 (1.0)	−0.5 (−0.7, −0.3) ***	3.7 (1.1)	3.6 (1.1)	0.1 (−0.1, 0.3)
Credibility ^b^	6.5 (3.2)	6.2 (2.8)	0.3 (−0.3, 0.9)	6.2 (2.3)	6.6 (3.0)	−0.4 (−1.0, 0.2)
Likeability ^c^	17.4 (4.6)	17.0 (3.9)	0.4 (−0.4, 1.3)	17.0 (4.4)	17.3 (4.1)	−0.3 (−1.2, 0.5)
Perceived effectiveness ^d^	6.6 (2.5)	6.4 (2.6)	0.2 (−0.3, 0.7)	6.3 (2.6)	6.6 (2.5)	−0.3 (−0.8, 0.2)

*** Significant at *p*-value < 0.0001; MD= mean difference; CI = confidence interval. ^a^ Valence was measured on a 5-point scale ranging from very negative (1) to very positive (5). ^b^ Credibility was a sum of 3 items measured on a 15-point scale ranging from describes message very well (3) to describes message very poorly (15). ^c^ Likability was a sum of 6 items measured on a 25-point scale ranging from describes message very well (6) to describes message very poorly (30). ^d^ Perceived effectiveness was a sum of 2 items measured on a 9-point scale ranging from strongly disagree (2) to strongly agree (10).

**Table 3 ijerph-21-00207-t003:** Comparison of mean scores for HAV uptake across message frame groups for health belief model questions.

Health Belief Model Measures	Loss Frame (n = 197)	Gain Frame (n = 182)	Loss vs. Gain MD (95% CI)	Control (n = 93)	Loss vs. Control MD (95% CI)	Gain vs. Control MD (95% CI)
Perceived susceptibility						
Perceived need ^a^	3.2 (1.3)	2.9 (1.4)	0.2 (0.0, 0.5)	3.4 (1.3)	−0.2 (−0.6, 0.1)	−0.5 (−0.8, −0.1) **
Perceived risk	3.2 (1.2)	3.0 (1.3)	0.2 (0.0, 0.5)	3.3 (1.3)	−0.1 (−0.4, 0.2)	−0.3 (−0.7, 0.0)
Perceived nenefits ^b^	8.4 (1.7)	8.1 (1.8)	0.3 (−0.1, 0.6)	8.2 (1.5)	0.2 (−0.2, 0.6)	0.0 (−0.5, 0.4)
Perceived narriers ^c^	9.3 (4.3)	9.5 (4.3)	−0.2 (−1.1, 0.7)	10.7 (4.4)	−1.3 (−2.5, −0.2) *	−1.2 (−2.3, 0.0) *
Cue to action ^d^	18.9 (4.0)	17.8 (4.3)	1.1 (0.2, 2.0) *	18.3 (3.9)	0.6 (−0.4, −1.7)	−0.4 (−1.6, 0.7)
	Collective Frame (n = 188)	Individual Frame (n = 191)	Collective vs. Individual MD (95% CI)	Control (n = 93)	Collective vs. Control MD (95% CI)	Individual vs. Control MD (95% CI)
Perceived susceptibility						
Perceived need	2.9 (1.3)	3.1 (1.3)	−0.2 (−0.5, 0.1)	3.4 (1.3)	−0.5 (−0.8, −0.1) **	−0.3 (−0.6, 0.1)
Perceived risk	3.1 (1.2)	3.2 (1.3)	−0.1 (−0.4, 0.1)	3.3 (1.3)	−0.3 (−0.6, 0.0)	−0.1 (−0.5, 0.2)
Perceived benefits ^b^	8.2 (1.9)	8.3 (1.6)	−0.1 (−0.5, 0.3)	8.2 (1.5)	0.0 (−0.4, 0.5)	0.1 (−0.3, 0.5)
Perceived barriers ^c^	9.1 (4.4)	9.8 (4.2)	−0.7 (−1.6, 0.2)	10.7 (4.4)	−1.6 (−2.7, −0.5) **	−0.9 (−2.0, 0.2)
Cue to action ^d^	18.6 (4.1)	18.2 (4.3)	0.4 (−0.5, 1.3)	18.3 (3.9)	0.3 (−0.7, 1.4)	−0.1 (−1.2, 1.0)

* Significant at *p*-value < 0.05, ** significant at *p*-value < 0.01; HAV = hepatitis A virus; MD = mean difference; CI = confidence interval. ^a^ Perceived need was not calculated for individuals who responded “not applicable” (n = 65) and were ineligible to answer this question. Perceived risk was not calculated for individuals who responded “not applicable” (n = 42) and were ineligible to answer this question. ^b^ Perceived benefits were a sum of 2 items measured on a 9-point scale ranging from strongly disagree (2) to strongly agree (10). Benefits were not calculated for individuals who responded “not applicable” (n = 37) and were ineligible to answer this question. ^c^ Perceived barriers were a sum of 4 items measured on a 16-point scale ranging from strongly disagree (4) to strongly agree (20). Barriers were not calculated for individuals who responded “not applicable” (n = 40) and were ineligible to answer this question. ^d^ Cue to action was a sum of 5 items measured on a 20-point scale ranging from strongly disagree (5) to strongly agree (25). Cue to action was not calculated for individuals who responded “not applicable” (n = 68) and were ineligible to answer this question.

**Table 4 ijerph-21-00207-t004:** Health behavior construct reliability as measured by survey questions.

Construct	Number of Items	Reliability	Example Question
Message credibility	3	0.881	“Please indicate to which degree the following adjectives describe the tweet—Believable”
Message likability	6	0.706	“Please indicate to which degree the following adjectives describe the tweet—Enjoyable”
Message perceived effectiveness	2	0.916	“Persuade someone to get the hepatitis A shot when it is recommended”
Health belief model: benefits	2	0.551	“The Hepatitis A shot can prevent me from getting Hepatitis A”
Health belief model: barriers	4	0.866	“The Hepatitis A shot is dangerous to my health”
Health belief model: cue to action	5	0.747	“I plan to or have already gotten the Hepatitis A shot this year“
Health belief model: susceptibility	2	0.754	“I have had the Hepatitis A shot before so I am no longer at risk”

## Data Availability

The data presented in this study are available on request from the corresponding author.

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
