# Peer review of "The Association between Message Framing and Intention to Vaccinate Predictive of Hepatitis A Vaccine Uptake"

_ijerph, 2024, doi:10.3390/ijerph21020207_

Round 1
Reviewer 1 Report
Comments and Suggestions for Authors
The following manuscript titled "The Association Between Message Framing and Intention to 2 Vaccinate Predictive of Hepatitis A Vaccine Uptake' evaluates the use of framing messages to understand its impact and intend on the population to get vaccinated. The article also uses different messaging strategies to evaluate its effect. Moreover, the study compares different messaging strategies to no strategy control group. The final conclusion of the study is that there is no effect of different messaging strategies compared to no message control group. Whereas the manuscript is well written and the groups well defined, there are some limitations to this study.
1. Can the authors comment of how this would impact in a different ethnic population? This study was conducted in a majority white and educated population, would the results differ if the population was underserved or low socio-economic?
2. What is the power of the study?
3. How would the messaging through news or other media would be taken by the population?
4. How would this translate to older people?
Author Response
- Can the authors comment of how this would impact in a different ethnic population? This study was conducted in a majority white and educated population, would the results differ if the population was underserved or low socio-economic?
We thank the reviewer for these insightful questions. We have added two sentences to the strengths and limitations paragraph discussing the limitations of having a majority white and educated population and highlighting the need for more research examining HAV vaccine intention in disadvantaged and vulnerable populations.
“Fourth, the majority of the study sample was young, White, and had a high level of education compared to the general US population. Therefore, more research is needed to examine the effects of these message frames among other populations vulnerable to HAV outbreaks that may not be available for research via MTurk, such as older adults.”
- What is the power of the study?
We thank the reviewer for this question and have added a paragraph about our post-hoc power analysis to the end of the methods section.
“We conducted a post-hoc power analysis to determine whether the existing sample size had adequate study power for all the proposed analytical aims. Assuming an effect size (Cohen’s d) = 0.5, α = 0.05, a sample size of 90 for group 1, and a sample size of 92 for group 2, we used a t-test to compute the achieved power for the difference in means between two independent groups. Our computations noted achieving 95.7% power. All power calculations were conducted using G*Power version 3.1.”
- How would the messaging through news or other media would be taken by the population?
We thank the reviewer for this question, however we currently do not have information about the uptake of messaging through news or other media. Therefore, we have a sentence in our conclusion stating that future research should consider examining use of different modalities, such as news, and other forms of social media on delivery of HAV-framing messages.
“Future research may also consider examining use of different modalities and other forms of social media on delivery of HAV-framing messages that may affect HAV vaccination.”
- How would this translate to older people?
Within our sentence in the discussion section highlighting the need for more research examining HAV vaccine intention in disadvantaged and vulnerable populations, we specify that older adults are a vulnerable population.
“Therefore, more research is needed to examine the effects of these message frames among other populations vulnerable to HAV outbreaks that may not be available for research via MTurk, such as older adults.”
Reviewer 2 Report
Comments and Suggestions for Authors
The article is well-structured and well-written. The experiments seem suitable to demonstrate their objectives. It tackles the topic of vaccination and aims to examine how the message influences both behavior and the self-reported intention to receive the HAV vaccine, as well as the behavioral determinants of intention.
Please consider to summarize the introduction, focusing on your aims.
Follow PRISMA flowchart and provide a PRISMA statement.
Did you register it on PROSPERO?
Results didn't conduct to addressing you hypothesis: what is the novelty?
Author Response
Please consider to summarize the introduction, focusing on your aims.
Thank you for this suggestion. We have moved the paragraph summarizing our aims to the end of the introduction sentence and added some additional details to clarify our aims.
“Furthermore, we included a no-message control group in our study design to assess whether not receiving any message influences self-reported intention to receive the HAV vaccine.”
Follow PRISMA flowchart and provide a PRISMA statement.
We believe a PRISMA flowchart and PRISMA statement are not applicable for this study because it was not a systematic review.
Did you register it on PROSPERO?
We did not register this study on PROSPERO because the study was not a systematic review.
Results didn't conduct to addressing you hypothesis: what is the novelty?
Thank you for this question. We have changed some of the wording in the third paragraph of our discussion section to highlight that this is the first study to date that has examined the effects of message framing in combination with a control group on HAV vaccine intention.
“Although prior work has looked at vaccine intention for diseases such as HPV, influen-za, and COVID-19, this is the first study to date that has examined the effects of mes-sage framing in combination with a control group on HAV vaccine intention.”
Reviewer 3 Report
Comments and Suggestions for Authors
The abstract nicely incorporated all major key points and messages to the readers.
The introduction is also well-written, but it would also be nice to point out the general acceptance of traditional vaccines such as HAV vaccines and new vaccines such as COVID-19. The vaccine hesitancy effect has a different background but influence from one to other exists.
Materials and methods are explained in detail.
There should be explained limitations of the study because there are only 2 participants with less than high school.
Disadvantaged and vulnerable groups were not mentioned and should be one of the focuses of the HAV vaccination population.
The authors should correct referencing according to the instructions to the authors.
Author Response
The introduction is also well-written, but it would also be nice to point out the general acceptance of traditional vaccines such as HAV vaccines and new vaccines such as COVID-19. The vaccine hesitancy effect has a different background but influence from one to other exists.
We thank the reviewer for this comment and have added a sentence to the end of our second paragraph in the introduction to discuss the influences of vaccine hesitancy and acceptance on new and traditional vaccines.
“While traditional vaccines, such as the HAV vaccine, and new vaccines, such as the COVID-19 vaccine, share the common goal of preventing infectious diseases, differences in their development, public perception, and the influence of vaccine hesitancy necessitate tailored communication strategies, such as message framing, to ensure widespread acceptance and participation in vaccination programs.”
There should be explained limitations of the study because there are only 2 participants with less than high school.
We thank the reviewer for this suggestion. We have added a sentence to the strengths and limitations paragraph discussing the limitations of having a highly educated study sample.
“Fourth, the majority of the study sample was young, White, and had a high level of education compared to the general US population.”
Disadvantaged and vulnerable groups were not mentioned and should be one of the focuses of the HAV vaccination population.
We thank the reviewer for this comment. We have added a sentence to the strengths and limitations paragraph highlighting the need for more research examining HAV vaccine intention in disadvantaged and vulnerable populations.
“Therefore, more research is needed to examine the effects of these message frames among other populations vulnerable to HAV outbreaks that may not be available for research via MTurk, such as older adults.”
The authors should correct referencing according to the instructions to the authors.
Thank you for bringing this to our attention. References have been corrected in the manuscript as the editorial office has made formatting changes to our original submission.